# Planning with General Objective Functions: Going Beyond Total Rewards

**Ruosong Wang**[*]
Carnegie Mellon University
ruosongw@andrew.cmu.edu

**Peilin Zhong**[*]
Columbia University
pz2225@columbia.edu

**Simon S. Du**
University of Washington, Seattle
ssdu@cs.washington.edu

**Ruslan Salakhutdinov**
Carnegie Mellon University
rsalakhu@cs.cmu.edu

**Lin F. Yang**
University of California, Los Angeles
linyang@ee.ucla.edu

## Abstract

Standard sequential decision-making paradigms aim to maximize the cumulative reward when interacting with the unknown environment., i.e., maximize $\sum_{h=1}^{H} r_h$ where $H$ is the planning horizon. However, this paradigm fails to model important practical applications, e.g., safe control that aims to maximize the lowest reward, i.e., maximize $\min_{h=1}^{H} r_h$. In this paper, based on techniques in sketching algorithms, we propose a novel planning algorithm in deterministic systems which deals with a large class of objective functions of the form $f(r_1, r_2, ...r_H)$ that are of interest to practical applications. We show that efficient planning is possible if $f$ is symmetric under permutation of coordinates and satisfies certain technical conditions. Complementing our algorithm, we further prove that removing any of the conditions will make the problem intractable in the worst case and thus demonstrate the necessity of our conditions.

## 1 Introduction

Markov decision process (MDP) is arguably the most popular model for sequential decision-making problems. MDP assumes both the transition function $T : \mathcal{S} \times \mathcal{A} \to \mathcal{S}$ and the reward function $r : \mathcal{S} \times \mathcal{A} \to \mathbb{R}$ only depend on the current state-action pair where $\mathcal{S}$ is the state space and $\mathcal{A}$ is the action space, and the objective of the agent is to maximize the summation of all rewards $\sum_{h=1}^{H} r_h$ where $H$ is the planning horizon and $r_h = r(s_h, a_h)$.

The drawback of the standard MDP model is that it even fails to capture some simple sequential decision-making tasks. For example, in self-driving, the goal is not to maximize the total reward but to maximize the *minimum* reward on the trajectory, say if one models a car crash as $-1$ reward and $0$ reward otherwise. Note that in this simple example, the state transition function $T$ and the reward function $r$ still satisfy the Markov property. The only difference is that the objective changes from maximizing the *sum* of rewards $\sum_{r=1}^{H} r_h$ to maximizing the *minimum* of rewards $\min_{h=1}^{H} r_h$.

This "small" difference requires the agent to change the planning strategy significantly because the agent needs to look at the full history of rewards. This gives rise the following natural problem:

> *Can we design a provably efficient algorithm for **general** objective functions?*

Here by efficient, we mean the complexity of the algorithm does not scale exponentially in $H$. This is a challenging question as existing approaches for MDP models cannot be applied here.

---

[*]Equal contribution

In this paper, we give a positive answer to the above question by designing an efficient algorithm for objective functions $f(r_1, r_2, \ldots, r_H)$ that satisfy certain technical conditions. Below we list several motivating examples of objective functions that satisfy these conditions.

1. $f(r_1, r_2, \ldots, r_H) = \min\{r_1, r_2, \ldots, r_h\}$: this objective function naturally formalizes sequential decision-making problems related to safety concerns, which we have discussed above.

2. $f(r_1, r_2, \ldots, r_H) = \max\{r_1, r_2, \ldots, r_h\}$: this objective function models the maximum reward-oriented behavior, which has been explicitly studied in the reinforcement learning literature, e.g., in [41], where the authors used this objective function to model certain financial problems.

3. $f(r_1, r_2, \ldots, r_H) = \text{median}\{r_1, r_2, \ldots, r_h\}$: maximizing cumulative rewards is equivalent to maximizing the mean of the reward values, which is not robust to adversarial perturbations and outliers. Maximizing the median or other quantiles of the reward values is a much more robust objective function, which is often used in situations where one seeks a robust solution. For instance, if each reward is collected by a noisy sensor, the median objective gives a much more robust solution than the mean objective.

4. $f(r_1, r_2, \ldots, r_H) = \sum_{k=1}^{K} r_{(k)}$ where $r_{(k)}$ represents the $k$-th largest reward in $\{r_h\}_{h=1}^{H}$: this objective function naturally models problems where the agent has a capacity constraint so that the agent can only keep the largest $K$ rewards.

Other objective functions have also appeared in previous work [47, 40, 34, 19, 37, 8, 39, 48, 13, 36]. We stress that the goal of this paper is not to study specific objective functions, but to give a characterization on the class of objective functions that admits provably efficient planning algorithms.

## 1.1 Our Contributions

In this paper, we develop an efficient algorithm that finds near-optimal policies in tabular deterministic systems for a wide range of objective functions. We assume there is an objective function $f : \mathbb{R}^H \to \mathbb{R}$, such that for a sequence of reward values $r_1, r_2, \ldots, r_H$, the objective function $f$ maps the reward values to an objective value $f(r_1, r_2, \ldots, r_H)$. Here $H$ is the planning horizon. We assume all reward values $r_h \in [0, 1]$ and the objective value $f(r_1, r_2, \ldots, r_H) \in [0, 1]$. Therefore, we may assume $f$ is a function that maps a vector in $[0, 1]^H$ to an objective value in $[0, 1]$.

We focus on the planning problem in tabular deterministic systems with general reward functions, i.e., given a deterministic system, our goal is to output a policy which (approximately) maximizes the objective function.[2] Before stating our results, we first give the three conditions on the objective function that our algorithm requires.

**Definition 1.1** (Symmetry). *For a function $f \in [0, 1]^H \to [0, 1]$, we say $f$ is* symmetric *if for any permutation $(i_1, i_2, \ldots, i_H)$ of $(1, 2, \ldots, H)$ and $x \in [0, 1]^H$, we have $f(x_1, x_2, \cdots, x_H) = f(x_{i_1}, x_{i_2}, \ldots, x_{i_H})$.*

**Definition 1.2** (Approximate Homogeneity). *Let $\bar{\varepsilon}, \bar{\delta} \in (0, 1)$. For a function $f \in [0, 1]^H \to [0, 1]$, we say $f$ satisfies $(\bar{\varepsilon}, \bar{\delta})$-approximate homogeneity if for any $x, y \in [0, 1]^H$ such that $x_h \in [y_h, (1 + \bar{\delta})y_h]$ for all $1 \leq h \leq H$, we have $f(y) \in [f(x) - \bar{\varepsilon}, f(x) + \bar{\varepsilon}]$.[3]*

**Definition 1.3** (Insensitivity to Small Entries). *Let $\hat{\varepsilon}, \hat{\delta} \in (0, 1)$. For a function $f \in [0, 1]^H \to [0, 1]$, we say $f$ is $(\hat{\varepsilon}, \hat{\delta})$-insensitive to small entires if for any $x \in [0, 1]^H$ we have $f(\overline{x}) \in [f(x) - \hat{\varepsilon}, f(x) + \hat{\varepsilon}]$, where $\overline{x}$ is a vector in $[0, 1]^H$ such that $\overline{x}_h = \begin{cases} x_h & \text{if } x_h \geq \hat{\delta} \\ 0 & \text{otherwise} \end{cases}$.*

Now we briefly discuss the three conditions that our algorithm requires. The first condition requires that the objective function $f$ is symmetric under permutation of coordinates. The second condition

requires that, for any input $x \in [0,1]^H$, if one increases each coordinate in $x$ multiplicatively by a factor of at most $(1 + \bar{\delta})$, then the error on the objective function $f$ is bounded by $\bar{\varepsilon}$. The final condition states that, for any input $x \in [0,1]^H$, truncating all entries smaller than $\hat{\delta}$ to zero leads to an approximation error of at most $\hat{\varepsilon}$. Given these conditions, now we state our main algorithmic result.

**Theorem 1.4** (Informal). *Given an objective function $f$ which is symmetric, $(\varepsilon/4, \hat{\delta})$-insensitive to small entries, and satisfies $(\varepsilon/4, \bar{\delta})$-approximate homogeneity, there is an algorithm that finds an $\varepsilon$-optimal policy in deterministic systems with time complexity $O((|\mathcal{S}||\mathcal{A}| + \mathcal{T}) \cdot H^{\Theta(\log(1/\hat{\delta})/\bar{\delta})})$ if evaluating the objective function $f$ on a single input costs $\mathcal{T}$ time.*

As stated in the theorem, the running time of our algorithm exponentially depends on $\log(1/\hat{\delta})/\bar{\delta}$. However, as we will show in examples given below, $\hat{\delta}$ and $\bar{\delta}$ are often constants if one aims at a policy with constant additive error, and therefore, our algorithm runs in polynomial time in those cases. Moreover, Our algorithm accesses the objective function $f$ in a black-box manner and thus automatically handles a large class of loss functions.

One may ask whether it is possible to remove those conditions in Definition 1.1-1.3. In Section 7, we further show that removing any of the three conditions will induce an exponential lower bound and makes the problem intractable in the worst-case. Therefore, all of our three conditions are necessary.

Below we give two large families of objective functions that can be handled by our algorithm. We note that these two families of objective functions have already included all examples mentioned in the introduction.

**Symmetric Norm.** A symmetric norm is a norm that satisfies the additional property that for any $x \in \mathbb{R}^H$, any permutation $\sigma$ and any assignment of $s_i \in \{-1, 1\}$, $f(x_1, x_2, \ldots, x_n) = f(s_1 x_{\sigma_1}, s_2 x_{\sigma_2}, \ldots, s_n x_{\sigma_n})$. Symmetric norm includes a large class of norms, for example the $\ell_p$ norm, the top-$k$ norm (the sum of absolute values of the leading $k$ coordinates of a vector), max-mix of $\ell_p$ norms (e.g. $\max\{\|x\|_2, c\|x\|_1\}$ for some $c > 0$), and sum-mix of $\ell_p$ norms (e.g. $\|x\|_2 + c\|x\|_1$ for some $c > 0$), as special cases. More complicated examples include the $k$-support norm [3] and the box-norm [35], which have found applications in sparse recovery.

For any symmetric norm $f$ that satisfies $f(x) \in [0,1]$ for any $x \in [0,1]^H$, $f$ is symmetric, $(\varepsilon, \varepsilon)$-insensitive to small entries and satisfies $(\varepsilon, \varepsilon)$-approximate homogeneity. Therefore, when applying our algorithm to such an objective function $f$, our algorithm finds an $\varepsilon$-optimal policy in time $O((|\mathcal{S}||\mathcal{A}| + \mathcal{T}) \cdot H^{\Theta(\log(1/\varepsilon)/\varepsilon)})$. Thus, our algorithm gives a polynomial-time approximation scheme (PTAS), i.e., the algorithm runs in polynomial time for any constant $\varepsilon > 0$.

**Lipschitz Functions.** Recall that a function $f : [0,1]^H \to [0,1]$ is Lipschitz continuous with respect to the $\ell_\infty$ norm with Lipschitz constant $L$ if for any $x, y \in \mathbb{R}^H$, $|f(x) - f(y)| \leq L\|x - y\|_\infty$. Clearly, such function $f$ is $(\varepsilon, \varepsilon/L)$-insensitive to small entries and satisfies $(\varepsilon, \varepsilon/L)$-approximate homogeneity. If $f$ is additionally symmetric, then our algorithm finds an $\varepsilon$-optimal policy in time $O((|\mathcal{S}||\mathcal{A}| + \mathcal{T}) \cdot H^{\Theta(\log(L/\varepsilon)L/\varepsilon)})$. Therefore, for constant $L$ and $\varepsilon$, our algorithm runs in polynomial time. An important example that satisfies the above conditions is the median function (or the $k$-th largest reward for any $k$), where we have $L = 1$ and thus our algorithm gives a PTAS.

## 2 Related Work

Most planning and reinforcement learning algorithms with provable guarantees rely on the MDP model. For the setting where the number of state and actions is finite, a.k.a. the tabular setting, considered in this paper, this is a long line of work trying to obtain the tight sample complexity and regret bounds [30, 45, 4, 1, 24, 28, 26]. Recently, there are attempts to generalize the tabular setting to more complicated scenarios [51, 17, 31, 25, 14, 46, 15, 27, 53, 38, 16]. However, to our knowledge, all these works only study the case where the objective function is the sum of total rewards and cannot be applied to the general objective functions considered in this paper. The only exception we are aware of is the work by [41], who studied the objective function $f(r_1, r_2, \ldots, r_H) = \max_{h=1}^H r_h$. However, the algorithm in [41] can not be applied to the general class of objective functions.

In our algorithm we adopt the layering technique first proposed by [23]. Such technique has been widely applied in the streaming and sketching literature. We refer interested readers to [10, 29, 2, 9, 7]

and references therein. However, to our best knowledge, this is the first time that the layering technique appears in sequential decision-making algorithms.

There is a line of research studying empirical aspects of non-Markovian rewards [5, 49]. See [49] for early schemes. These works define a task specification model and then generate a reward function to fulfill that specification. Classical models are based on sub-goal sequences [42, 43] and linear temporal logic and its variants [44, 18, 32, 11, 33, 50, 20], while recent approaches include reward machines [21, 12, 52, 22]. However, these works focus on empirical aspects of RL with non-Markovian rewards, while we design the first provably efficient algorithm under a set of necessary and sufficient conditions on the objective function in this work.

Another line of research considers risk-sensitive optimization in reinforcement learning. Examples of objectives that are considered in this line of search include a mean-variance criterion [47, 40, 34, 19], Conditional Value at Risk (CVaR) [37, 8, 39, 48, 13] and a Chernoff functional [36]. These papers focus on specific risk-sensitive objectives, while in this work we give a provably-efficient algorithmic framework to handle a large class of reward functions.

# 3 Preliminaries

**Notations.** Throughout the paper, for a positive integer $H$, we use $[H]$ to denote the set $\{1, 2, \ldots, H\}$. We use $\|x\|_p$ to denote the $\ell_p$ norm of a vector $x$. For a condition $\mathcal{E}$, we use $\mathbb{1}[\mathcal{E}]$ to denote the indicator function, i.e., $\mathbb{1}[\mathcal{E}] = 1$ if $\mathcal{E}$ holds and $\mathbb{1}[\mathcal{E}] = 0$ otherwise.

**Deterministic Systems.** Let $\mathcal{D} = (\mathcal{S}, \mathcal{A}, T, r, H, f)$ be a deterministic system, where $\mathcal{S}$ is the set of states, $\mathcal{A}$ is the set of actions, $T : \mathcal{S} \times \mathcal{A} \to \mathcal{S}$ is a deterministic transition function, $r : \mathcal{S} \times \mathcal{A} \to [0, 1]$ is a reward function,[4] $H \in \mathbb{Z}_+$ is the planning horizon, and $f$ is a objective function $f : [0, 1]^H \to [0, 1]$.[5] We assume there is a fixed initial state $s_1 \in \mathcal{S}$ in the deterministic system.

A policy $\pi$ chooses an action $a$ based on the current state $s \in \mathcal{S}$ and the time step $h \in [H]$. Formally, $\pi = \{\pi_h\}_{h=1}^H$ where for each $h \in [H]$, $\pi_h : \mathcal{S} \to \mathcal{A}$ maps a given state to an action. The policy $\pi$ induces a trajectory $s_1, a_1, r_1, s_2, a_2, r_2, \ldots, s_H, a_H, r_H$, where $a_1 = \pi_1(s_1)$, $r_1 = r(s_1, a_1)$, $s_2 = T(s_1, a_1)$, $a_2 = \pi_2(s_2)$, $r_2 = r(s_2, a_2)$, etc.

In this paper, we focus on the planning problem in deterministic systems with a general class of objective functions. Given a deterministic system $\mathcal{D}$, our goal is to efficiently find a policy $\pi$ that (approximately) maximizes the objective value $f(\pi) = f(r_1, r_2, \ldots, r_H)$. We use $\pi^*$ to denote the optimal policy and $f(\pi^*)$ to denote its objective value. We say a policy $\pi$ is $\varepsilon$-optimal if $f(\pi) \geq f(\pi^*) - \varepsilon$.

# 4 Algorithm for the $k$-th Largest Reward

As a warm-up, we consider a simple case where the objective function is the $k$-th largest value among the rewards collected along the trajectory, i.e., $f : [0, 1]^H \to [0, 1]$ is defined as $x \in [0, 1]^H$, $f(x) = x_{i_k}$, where $(i_1, i_2, \cdots, i_H)$ is a permutation of $(1, 2, \ldots, H)$ such that $x_{i_1} \geq x_{i_2} \geq \cdots \geq x_{i_H}$.

**High-level Ideas.** Despite being an important objective function, it is not immediately clear how to efficiently find a near-optimal policy for the objective function defined above. The main technical issue here is that one cannot simply use Bellman-type dynamic programming to solve this problem. Here, we reformulate the problem and develop a dynamic programming algorithm with an augmented state space to solve the reformulated problem. More specifically, finding a policy $\pi$ that maximizes the $k$-th largest reward is equivalent to finding the largest reward value $r$ such that there exists a policy $\pi$ so that there are at least $k$ rewards with value at least $r$ on the trajectory induced by $\pi$. Therefore, we enumerate the largest reward value $r$ and use a dynamic programming approach, which will be described in more detail below, to find a policy $\pi$ to maximize the number of rewards with value at least $r$ on the induced trajectory. To avoid enumerate a continuous reward value $r$, in our algorithm,

we first discretize reward values so that all reward values are in $\{0, \varepsilon, 2\varepsilon, \ldots, 1\}$ which induces an additive approximation error of at most $\varepsilon$.

**Dynamic Programming.** Here we give a detailed description of the dynamic programming algorithm with augmented state space mentioned above. For each state $s \in \mathcal{S}$, $l \in [1/\varepsilon]$ and a policy $\pi$, we use $V_h^\pi(s, l)$ to denote the number of discretized rewards with value at least $l \cdot \varepsilon$ for a trajectory starting from state $s$ induced by policy $\pi$ at level $h$, and $Q_h^\pi(s, a, l)$ to denote the number of discretized rewards with value at least $l \cdot \varepsilon$ for a trajectory starting from state-action pair $(s, a)$ induced by policy $\pi$ at level $h$. Similar to standard dynamic programming approaches for planning, we use $V_h^*(s, l)$ to denote the largest number of discretized rewards with value at least $l \cdot \varepsilon$ for a trajectory starting from state $s$ induced by any policy $\pi$ at level $h$, and $Q_h^*(s, a, l)$ is defined analogously. Note that $V^*(\cdot, \cdot)$ and $Q^*(\cdot, \cdot, \cdot)$ can be efficiently calculated using a Bellman-type dynamic programming approach. More specifically, for each $s \in \mathcal{S}$, $a \in \mathcal{A}$ and $l \in [1/\varepsilon]$, we have

$$V_h^*(s, l) = \max_{a \in \mathcal{A}} Q_h^*(s, a, l)$$

and

$$Q_h^*(s, a, l) = \begin{cases} \mathbb{1}[r(s, a) \geq l \cdot \varepsilon] & \text{if } h = H \\ V_{h+1}^*(T(s, a), l) + \mathbb{1}[r(s, a) \geq l \cdot \varepsilon] & \text{otherwise} \end{cases}.$$

Clearly, the above dynamic programming algorithm can be readily implemented in $O(|\mathcal{S}||\mathcal{A}|H/\varepsilon)$ time.

**Output the Policy.** In order to find a policy that approximately maximizes the $k$-th largest reward, we enumerate all possible $l \in [1/\varepsilon]$, and find the largest $l$ such that $V_1^*(s_1, l) \geq k$ where $s_1$ is the initial state. For such $l \in [1/\varepsilon]$, we know that there exists a policy so that on the induced trajectory, there are at least $k$ rewards with value at least $l \cdot \varepsilon$. Moreover, for any $l' > l$, there exists no policy such that there are at least $k$ rewards with value at least $l' \cdot \varepsilon$. These two facts imply that $l \cdot \varepsilon$ is indeed the best result one can achieve, up to an additive approximation error of at most $\varepsilon$.

It is also easy to output a policy using $V^*(\cdot, \cdot)$ and $Q^*(\cdot, \cdot, \cdot)$ obtained by the dynamic programming algorithm. To output a policy, we may choose actions greedily with respect to $Q^*(\cdot, \cdot, l)$. Formally, we define the policy to be $\pi_h(s) = \operatorname{argmax}_{a \in \mathcal{A}} Q_h^*(s, a, l)$. Using such a policy, we are effectively choosing a policy to maximize the number of rewards with value at least $l \cdot \varepsilon$ on the induced trajectory. On the other hand, since $V_1^*(s_1, l) \geq k$, it is guaranteed that there at least $k$ rewards with value at least $l \cdot \varepsilon$ on some trajectory, and thus the $k$-th largest reward is at least $l \cdot \varepsilon$.

Before getting into more complicated algorithms for more general cases, we would like to review the key ideas in our approach. For an objective function like the $k$-th largest reward which globally depends on all reward values on the trajectory, we show that it is possible to keep using the Bellman-like dynamic programming approach if one reformulates the problem carefully and augments the state space. Such ideas will be a crucial part of our final algorithm for solving the general case.

## 5 Algorithm for General Symmetric Norms

In this section, we consider a more general case where the objective function $f : [0, 1]^H \to [0, 1]$ is a symmetric norm. Recall that a symmetric norm $f(\cdot)$ is a norm and satisfies the additional property that for any $x \in \mathbb{R}^H$, any permutation $\sigma$ and any assignment of $s_i \in \{-1, 1\}$,

$$f(x_1, x_2, \ldots, x_n) = f(s_1 x_{\sigma_1}, s_2 x_{\sigma_2}, \ldots, s_n x_{\sigma_n}).$$

**High-level Ideas.** Similar to the previous algorithm, in the new one, we still discretize the reward values. More specifically, for each reward value, we discretize it to its nearest value in $\{1, 1/2, 1/4, \ldots, \varepsilon\}$, and we truncate all reward values less than $\varepsilon$ to zero. Compared to the approach in Section 4, the advantage of this new approach is that there are only $O(\log(1/\varepsilon))$ different discretized reward values. Since the approximation ratio of our algorithm depends on the number of different discretized reward values, this new discretization approach gives a much better approximation ratio. Our algorithm is built upon the layering framework in the sketching literature [23, 7]. For a symmetric norm, its value is completely determined by the histogram of the input, and we approximate the histogram by discretizing the input reward values. Here, the main observation is

that, after the discretization, there exists a "contributing reward value" such that even if we set all other rewards values to be zero and find an optimal policy with respect to this specific reward value, the resulting policy will still be a good approximation to the original problem. Moreover, finding the optimal policy for a specific reward is equivalent to finding a policy with maximum number of non-zero reward values, which can be efficiently solved using dynamic programming.

In our algorithm, we enumerate all discretized reward values $r$ in $\{1, 1/2, 1/4, \ldots, \varepsilon\}$. For each $r$, for all state-action pair $(s, a)$ whose discretized reward value does not equal $r$, we change $r(s, a)$ to $0$. After this step, the new discretized reward values will be binary, i.e., all discretized reward values will either be $r$ or $0$, and then we find a policy to maximize the symmetric norm objective function for the deterministic system with binary reward values. Since there are $\log(1/\varepsilon) + 1$ possible values for $r$, effectively we find $\log(1/\varepsilon) + 1$ policies during our algorithm. We return the policy with largest objective function value. In the remaining part of this section, we first show how to find a policy to maximize a symmetric norm objective function in a deterministic system with binary reward values, and then analyze the approximation ratio of our algorithm.

To proceed, we need the following fact regarding symmetric norms.

**Lemma 5.1** (Proposition IV.1.1 in [6]). *If $f(\cdot)$ is a symmetric norm, for any $x, y \in \mathbb{R}^H$ such that $|x_i| \leq |y_i|$ for all $i$, we have $f(x) \leq f(y)$.*

The above lemma implies that symmetric norms satisfy $(\varepsilon, \varepsilon)$-approximate homogeneity for any $\varepsilon > 0$. Now we are ready to present the algorithm for general symmetric norms.

**Dynamic Programming.** For a deterministic system with binary reward values, in order to maximize a symmetric norm objective function, we only need to maximize the number of non-zero reward values on the trajectory, which follows from the monotonicity property of symmetric norms in Lemma 5.1. In order to find a policy that maximizes the number of non-zero reward values on the trajectory, we may use an approach similar to the classical Bellman-type dynamic programming. For each state $s \in \mathcal{S}$ and policy $\pi$, we use $V_h^\pi(s)$ to denote the number of non-zero reward values for a trajectory starting from state $s$ induced by policy $\pi$ at level $h$, and $Q_h^\pi(s, a)$ is defined similarly. We define $V_h^*(s)$ to be the largest number of non-zero reward values for a trajectory starting from $s$ induced by any policy at level $h$, and $Q^*(s, a)$ is defined analogously. We note that $V^*(\cdot)$ and $Q^*(\cdot, \cdot)$ can be efficiently calculated, since

$$V_h^*(s) = \max_{a \in \mathcal{A}} Q_h^*(s, a)$$

and

$$Q_h^*(s, a) = \begin{cases} \mathbb{1}[r(s, a) > 0] & \text{if } h = H \\ V_{h+1}^*(T(s, a)) + \mathbb{1}[r(s, a) > 0] & \text{otherwise} \end{cases}.$$

**Approximation Ratio.** Now it remains to analyze the approximation ratio of our algorithm. First, we truncate all reward values less than $\varepsilon$ to zero, and thus an additive approximation error of $\varepsilon$ will be induced. Moreover, reducing each reward value to its nearest value in $\{1, 1/2, 1/4, \ldots, \varepsilon\}$ will induce a multiplicative approximation error of at most $2$. From now on, we assume all reward values are in $\{1, 1/2, 1/4, \ldots, \varepsilon, 0\}$. Suppose that the discretized reward values on the trajectory induced by the optimal policy is $r = (r_1, r_2, \ldots, r_H)$. For each $0 \leq i \leq \log(1/\varepsilon)$, we define a new vector $\hat{r}^{(i)}$ whose $h$-th entry is $r_h$ if $r_h$ is exactly $2^{-i}$ and $0$ otherwise. An important observation is that

$$\max_i f(\hat{r}^{(i)}) \geq \Omega(f(r)/\log(1/\varepsilon)),$$

since by triangle inequality we have

$$\sum_{i=0}^{\log(1/\varepsilon)} f(\hat{r}^{(i)}) \geq f(r).$$

Moreover, in our algorithm, for each value in $\{1, 1/2, 1/4, \ldots, \varepsilon\}$, we find a policy which maximizes the number of rewards with that value on the trajectory induced by the policy, which means the objective value of the found policy is at least

$$\max_i f(\hat{r}^{(i)}) \geq \Omega(f(r)/\log(1/\varepsilon)).$$

Combining with the truncation step that removes all reward values less than $\varepsilon$, our algorithm is guaranteed to output a policy with objective value at least $\Omega((f(\pi^*) - \varepsilon)/\log(1/\varepsilon))$, where $f(\pi^*)$ is the objective value of the optimal policy. We remark that the approximation guarantee of this algorithm can be improved by using more sophisticated discretization procedure. Here we give this algorithm simply for motivating algorithms in later sections, and thus do not focus on optimizing the approximation guarantee.

# 6 Algorithm for General Objective Functions

In this section, we present our algorithm for finding $\varepsilon$-optimal policies for deterministic systems with general reward functions. We assume the objective function $f$ is symmetric, $(\varepsilon/4, \hat{\delta})$-insensitive to small entries, and satisfies $(\varepsilon/4, \bar{\delta})$-approximate homogeneity. We first give the high-level ideas of our algorithm. The formal description is given in Algorithm 1, and we give the formal analysis of our algorithm in the supplementary material.

**High-level Ideas.** Our algorithm combines ideas in Section 4 and Section 5. We discretize the reward values using a similar approach as in Section 5, and then find an optimal policy for the discretized reward values using dynamic programming with an augmented state space. Below we give more details for these two main components of our algorithm.

**Discretization.** Similar to Section 5, we discretize reward values so that all reward values are in $\{\hat{\delta}, \hat{\delta} \cdot (1 + \bar{\delta}), \hat{\delta} \cdot (1 + \bar{\delta})^2, \ldots\}$, and truncate all reward values less than $\hat{\delta}$ to zero. Formally, for a state-action pair $(s, a)$, the discretized reward value $\hat{r}(s, a)$ is defined as

$$\hat{r}(s, a) = \begin{cases} 0 & r(s, a) < \hat{\delta} \\ \hat{\delta} \cdot (1 + \bar{\delta})^j & r(s, a) \in [\hat{\delta} \cdot (1 + \bar{\delta})^j, \hat{\delta} \cdot (1 + \bar{\delta})^{j+1}) \end{cases}.$$

There are two advantages of using such a discretization approach. First, there are only $\log_{1+\bar{\delta}}(1/\hat{\delta}) = \Theta(\log(1/\hat{\delta})/\bar{\delta})$ different reward values after discretization. Since the running time of our dynamic programming algorithm depends exponentially on the number of different reward values, such a discretization approach significantly improves the efficiency of our algorithm. Moreover, since the reward function $f$ is $(\varepsilon/4, \hat{\delta})$-insensitive to small entries and satisfies $(\varepsilon/4, \bar{\delta})$-approximate homogeneity, the additive error induced by the discretization approach is upper bounded by $\varepsilon/2$. Therefore, we can find an $\varepsilon$-optimal policy for the original problem if we can find an optimal policy for the deterministic system with discretized reward values.

**Dynamic Programming.** After the discretization step, the state space for possible reward values has been significantly reduced, and we use a dynamic programming approach to find the optimal policy. For a policy $\pi$ and a state $s \in \mathcal{S}$, we use $V_h^\pi(s)$ to denote the multiset of reward values on the trajectory starting from state $s$ induced by policy $\pi$ at level $h$. We use $V_h^*(s)$ to denote the set of all possible multisets of reward values on trajectories induced by all policies at level $h$, i.e., $V_h^*(s) = \cup_\pi \{V_h^\pi(s)\}$. $Q_h^\pi(s, a)$ and $Q_h^*(s, a)$ are defined analogously. Here, we may safely ignore the order of the reward values since the objective function $f$ is symmetric. Moreover, for each $s \in \mathcal{S}$, the size of $V_h^*(s)$ is upper bounded by $H^{\Theta(\log(1/\hat{\delta})/\bar{\delta})}$, since for each discretized reward value $r$, there are at most $H$ rewards with discretized value $r$ on a trajectory, and there are only $\log_{1+\bar{\delta}}(1/\hat{\delta}) = \Theta(\log(1/\hat{\delta})/\bar{\delta})$ different reward values after the discretization.[6] As shown in Algoriithm 1, $V^*(\cdot)$ and $Q^*(\cdot, \cdot)$ can be efficiently calculated, using a Bellman-type dynamic programming algorithm.

**Output the Policy.** In order to find a policy for the discretized reward values, we enumerate all multisets of reward values $\mathcal{R} \in V_1^*(s_1)$, and find the one with the largest objective value $f(\mathcal{R})$. In order to output the policy, we start from the initial state $s_1$, find an action $a \in \mathcal{A}$ such that $\mathcal{R} \in Q^*(s, a)$, remove $\hat{r}(s, a)$ from $\mathcal{R}$ and continue this procedure inductively.

**Running Time.** As mentioned above, for each state-action pair $(s, a)$, the size of $Q_h^*(s, a)$ and $V_h^*(s)$ is upper bounded by $H^{\Theta(\log(1/\hat{\delta})/\bar{\delta})}$. Therefore, the running time of the dynamic programming part is at most $O(|\mathcal{S}||\mathcal{A}| \cdot H^{\Theta(\log(1/\hat{\delta})/\bar{\delta})})$. Moreover, in order to output the policy, we evaluate the objective function $f$ on $H^{\Theta(\log(1/\hat{\delta})/\bar{\delta})}$ different inputs. Suppose evaluating the objective function $f$ on a single input costs $\mathcal{T}$ time, the total running time of our algorithm will be $O((|\mathcal{S}||\mathcal{A}| + \mathcal{T}) \cdot H^{\Theta(\log(1/\hat{\delta})/\bar{\delta})})$.

**Approximation Guarantee.** Under the assumption that the objective function $f$ is symmetric, $(\varepsilon/4, \hat{\delta})$-insensitive to small entries, and satisfies $(\varepsilon/4, \bar{\delta})$-approximate homogeneity, for any vector $r \in [0, 1]^H$, we have $|f(r_1, r_2, \ldots, r_H) - f(\hat{r}_1, \hat{r}_2, \ldots, \hat{r}_H)| \leq \varepsilon/2$, where for each $h \in [H]$, $\hat{r}_h$ is the discretized value of $r_h$ as defined above. Since our algorithm finds the optimal policy with respect to the discretized reward values, the policy $\pi$ returned by our algorithm satisfies $f(\pi) \geq f(\pi^*) - \varepsilon$.

---

**Algorithm 1** Deterministic Systems with General Reward Functions
___
1: **for** $h \in [H]$ **do**
2:     **for** $(s, a) \in \mathcal{S}_h \times \mathcal{A}$ **do**
3:         Let $Q^*(s, a) = \begin{cases} \{\{\hat{r}(s, a)\}\} & \text{if } h = H \\ \{\mathcal{R} \cup \{\hat{r}(s, a)\} \mid \mathcal{R} \in V^*(T(s, a))\} & \text{otherwise} \end{cases}$
4:         Let $V^*(s) = \cup_{a \in \mathcal{A}} Q^*(s, a)$.
5:     **end for**
6: **end for**
7: Initialize policy $\bar{\pi}$ arbitrarily.
8: **for** $\mathcal{R} \in V^*(s_1)$ **do**
9:     Initialize policy $\pi_{\mathcal{R}}$ arbitrarily.
10:     Let $\mathcal{R}^1 = \mathcal{R}$.
11:     **for** $h \in [H]$ **do**
12:         Let $\pi_{\mathcal{R}}(s_h) = a \in \mathcal{A}$ such that $\mathcal{R}^h \in Q^*(s, a)$.
13:         Let $\mathcal{R}^{h+1} = \mathcal{R}^h \setminus \{\hat{r}(s_h, a)\}$ and $s_{h+1} = T(s_h, \pi_{\mathcal{R}}(s_h))$.
14:     **end for**
15:     Let $\bar{\pi} = \pi_{\mathcal{R}}$ if $f(\pi_{\mathcal{R}}) > f(\bar{\pi})$.
16: **end for**
17: Return $\bar{\pi}$.

---

## 7 Hardness Results

In this section, we prove that, without any of the three assumptions, any algorithm needs to query the values of $f$ for exponentially many different inputs vectors to find a near-optimal policy. More concretely, since we consider algorithms that can deal with a large family of objective functions, we assume that the algorithm access the objective function $f$ in a black-box manner, and we prove exponential lower bounds on the number of times that the algorithm evaluates the objective function $f$. Since the query complexity lower bounds the running time, our hardness results demonstrate that all of our three assumptions are necessary to ensure the intractability of the problem.

**Hard Instance.** In our hard instances, in each level $h \in [H]$, there is a single state $s_h \in \mathcal{S}_h$. There are two actions $a_1$ and $a_2$ in the action space $\mathcal{A}$, and $T(s_h, a_1) = T(s_h, a_2) = s_{h+1}$ for any $1 \leq h < H$.

**Necessity of Symmetry.** We first show that if the objective function $f$ is insensitive to small entries, satisfies approximate homogeneity but is not symmetric, then any algorithm still needs to query exponential number of values of $f$ to find a near-optimal policy, and thus demonstrate the necessity of the assumption that $f$ is symmetric. Here we have $r(s, a_1) = 1/2$ and $r(s, a_2) = 1$ for any $s \in \mathcal{S}$. Now we define the objective function $f$, which is parameterized by a vector $\theta \in \{1/2, 1\}^H$.

For a vector $\theta \in \{1/2, 1\}^H$, we define a function $f_\theta : [0, 1]^H \to [0, 1]$. For a vector $x \in [0, 1]^H$, if there exists $x_h = 0$ for some $h \in [H]$ then we define $f_\theta(x) = 0$. Otherwise,

$$f_\theta(x) = \min_{h \in [H]} \min\{x_h/\theta_h, \theta_h/x_h\}.$$

It is easy to verify that for any $\varepsilon > 0$, $f$ satisfies $(\varepsilon, \varepsilon)$-approximate homogeneity and is $(2\varepsilon, \varepsilon)$-insensitive to small entries. In the hard instance, the objective function $f$ is set to be $f_\theta$, where $\theta$ is one of the $2^H$ vectors in $\{1/2, 1\}^H$.

Recall that in our hard instance, all reward values are in $\{1/2, 1\}$, and for any $x \in \{1/2, 1\}^H$, $f_\theta(x) = 1$ if $x = \theta$, and $f_\theta(x) = 1/2$ if $x \neq \theta$. Therefore, to receive an objective value of 1, the agent must choose the correct actions for all the $H$ steps, and otherwise the agent will always receive an objective value of $1/2$. Here, the optimal policy is $\pi(s_h) = a_1$ if $\theta_h = 1/2$ and $\pi(s_h) = a_2$ if $\theta_h = 1$. Therefore, the correct actions are fully encoded in the vector $\theta$. However, there are $2^H$ possible vectors for $\theta$. Therefore, to find the correct actions for all the $H$ steps, the agent must enumerate all the $2^H$ possible combinations of actions to figure out the underlying vector $\theta$. This intuition is made formal is the supplementary material using Yao's minimax principle [54].

**Necessity of Approximate Homogeneity.** Here we show that if the objective function $f$ is symmetric, insensitive to small entries but does not satisfy approximate homogeneity, then any algorithm still needs to query exponential number of values of $f$ to find a near-optimal policy, and thus demonstrate the necessity of approximate homogeneity. Here we have $r(s_h, a_1) = (2h + 2H - 1)/4H$ and $r(s, a_2) = (h + H)/2H$ for any $h \in [H]$. Now we define the objective function $f$, which is parameterized by a vector $\theta \in \mathbb{R}^H$ where $\theta_h \in \{(2h + 2H - 1)/4H, (h + H)/2H\}$ for all $h \in [H]$.

For any vector $x \in [0, 1]^H$, we use $i_1, i_2, \ldots, i_H$ to denote a permutation of $(1, 2, \ldots, H)$ such that $0 \leq x_{i_1} \leq x_{i_2} \leq \ldots \leq x_{i_H} \leq 1$. We define $f_\theta(x) = 1$ if $x_{i_h} = \theta_h$ for all $h \in [H]$, and $f_\theta(x) = 0$ otherwise. Clearly, $f$ is symmetric and $(0, \varepsilon)$-insensitive to small entries for any $\varepsilon \leq 1/2$, but does not satisfy approximate homogeneity. The objective function $f$ is set to be $f_\theta$, where $\theta$ is one of the $2^H$ vectors defined above. In order to receive an objective value of 1, the agent must choose the correct actions for all the $H$ steps, and otherwise the agent will always receive an objective value of 0, which implies an exponential query lower bound using the same argument mentioned above.

**Necessity of Insensitivity to Small Entries.** Here we show that if the objective function $f$ is symmetric, satisfies approximate homogeneity but is not insensitive to small entries, then any algorithm still needs to query exponential number of values of $f$ to find a near-optimal policy, and thus demonstrate the necessity of insensitivity to small entries. Here we have $r(s_h, a_1) = 2^{-H(2h-1)}$ and $r(s_h, a_2) = 2^{-2Hh}$ for any $h \in [H]$. Now we define the objective function $f$, which is parameterized by a vector $\theta \in \mathbb{R}^H$ where $\theta_h \in \{2^{-H(2h-1)}, 2^{-2Hh}\}$ for all $h \in [H]$.

For a vector $\theta$ satisfying the above condition, we define a function $f_\theta : [0, 1]^H \to [0, 1]$. For a vector $x \in [0, 1]^H$, if there exists $x_h = 0$ for some $h \in [H]$ then we define $f_\theta(x) = 0$. Otherwise, we use $i_1, i_2, \ldots, i_H$ to denote a permutation of $(1, 2, \ldots, H)$ such that $1 \geq x_{i_1} \geq x_{i_2} \geq \ldots \geq x_{i_H} \geq 0$, and we define

$$f_\theta(x) = \min_{h \in [H]} \min\{x_{i_h}/\theta_h, \theta_h/x_{i_h}\}.$$

For any $\varepsilon > 0$, $f$ satisfies $(\varepsilon, \varepsilon)$-approximate homogeneity. It is also clear that $f$ is symmetric. In the hard instance, the objective function $f$ is set to be $f_\theta$, where $\theta$ is one of the $2^H$ vectors defined above. To receive an objective value of 1, the agent must choose the correct actions for all the $H$ steps, and otherwise the agent will receive an objective value of $1/2$. The lower bound can be proved using the same argument as above.

# 8 Conclusion

In this paper, we study planning problems with general objective functions in deterministic systems, and give the first provably efficient algorithm for a broad class of objective functions that satisfy certain technical conditions. We complement our positive results by showing that these conditions are necessary. An interesting direction is to extend our results to stochastic environments. Another interesting future direction is to study sequential decision-making problems with a huge state space and a general objective function for which one needs to combine function approximation techniques with the analysis in our paper.

## Broader Impact

This work is mainly theoretical. By devising provably efficient algorithms for planning with general objective functions, we believe our various algorithmic insights (discretization, augmenting state space) could potentially guide practitioners to design efficient and theoretically-principled planning algorithms that work for various settings.

## Disclosure of Funding

Ruosong Wang and Ruslan Salakhutdinov were supported in part by NSF IIS1763562, US Army W911NF1920104 and ONR Grant N000141812861. Peilin Zhong is supported in part by NSF grants CCF-1740833, CCF-1703925, CCF-1714818 and CCF-1822809 and a Google Ph.D. Fellowship. Part of the work is done while Simon S. Du was at the Institute for Advanced Study where he was supported by NSF grant DMS-1638352 and the Infosys Membership.

## Footnotes

[2]We remark that in deterministic systems, the planning problem is almost equivalent to the learning problem (i.e., the agent needs to interact with the environment to learn the transition and the reward), since the agent can readily reach all state-action pairs and learn the transition and reward using linear number of samples.

[3]We remark that the condition $f(y) \in [f(x) - \bar{\varepsilon}, f(x) + \bar{\varepsilon}]$ can be changed to $f(y) \in [(1 - \bar{\varepsilon})f(x), (1 + \bar{\varepsilon})f(x)]$ so that the error on the objective function value is also multiplicative. Note that the later condition is strictly stronger since $f(x) \leq 1$ for any $x \in [0, 1]^H$.

[4] For the sake of representation, here we assume the rewards are in $[0, 1]$. Our algorithm can be readily generalized to handle rewards in a different range.

[5] Again, the assumption that the objective value is in $[0, 1]$ is only for sake of presentation.

[6]Recall that the number of multisets of cardinality $k$ with elements taken from a finite set of cardinality $n$ is at most $k^n$. Notice that "multisets" are different from "sequence", i.e., for multisets we do not care about orders of elements.

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
