[Supplementary Material]

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

# A    Proof of Theorem 1.4

Recall that for a reward value $r$, the discretized reward value $\hat{r}$ is defined as

$$\hat{r} = \begin{cases} 0 & r < \hat{\delta} \\ \hat{\delta} \cdot (1+\bar{\delta})^j & r \in [\hat{\delta} \cdot (1+\bar{\delta})^j, \hat{\delta} \cdot (1+\bar{\delta})^{j+1}) \end{cases}.$$

We restate Theorem 1.4 as follow.

**Theorem A.1.** *Given an objective function $f : [0,1]^H \to [0,1]$ which is symmetric, $(\varepsilon/4, \hat{\delta})$-insensitive to small entries, and satisfies $(\varepsilon/4, \bar{\delta})$-approximate homogeneity, Algorithm 1 finds an $\varepsilon$-optimal policy in deterministic systems with time complexity*

$$O((|\mathcal{S}||\mathcal{A}| + \mathcal{T}) \cdot H^{\Theta(\log(1/\hat{\delta})/\bar{\delta})})$$

*if evaluating the objective function $f$ of a single policy costs $\mathcal{T}$ time.*

*Proof.* Let us first consider the running time of Algorithm 1. For each state-action pair $(s,a) \in \mathcal{S} \times \mathcal{A}$ and $h \in [h]$, each element $\mathcal{R}$ in $Q_h^*(s,a)$ is a multiset of discretized reward values. Since we only have $\log(1/\hat{\delta})/\bar{\delta}$ different discretized reward values and the size of $\mathcal{R}$ is at most $H$, the size of $Q_h^*(s,a)$ is at most $H^{O(\log(1/\hat{\delta})/\bar{\delta})}$. Thus, the running time of the first loop is at most $|\mathcal{S}| \cdot |\mathcal{A}| \cdot H^{O(\log(1/\hat{\delta})/\bar{\delta})}$. Since the number of different multisets $\mathcal{R}$ is at most $H^{O(\log(1/\hat{\delta})/\bar{\delta})}$ and we need $H \cdot |\mathcal{A}| + \mathcal{T}$ time for each iteration of the second loop of Algorithm 1, we need $O(H \cdot |\mathcal{A}| + \mathcal{T}) \cdot H^{\Theta(\log(1/\hat{\delta})/\bar{\delta})}$ time to find $\bar{\pi}$. Thus, the total running time is $O((|\mathcal{S}||\mathcal{A}| + \mathcal{T}) \cdot H^{\Theta(\log(1/\hat{\delta})/\bar{\delta})})$.

Next, we prove the correctness of the algorithm.

**Claim A.2.** *For any state $s \in \mathcal{S}$ and $h \in [H]$, a multiset $\mathcal{R}$ belongs to $V_h^*(s)$ if and only if there is a trajectory starting from state $s$ at level $h$ whose multiset of discretized reward values is exactly $\mathcal{R}$.*

*Proof.* Suppose $\mathcal{R} \in V_h^*(s)$. We want to show that there is a trajectory starting from $s$ at level $h$ whose multiset of discretized reward values is exactly $\mathcal{R}$. The proof is by induction on $h \in [H]$. Consider the base case when $h = H$. Let $s_H \in \mathcal{S}_H$. We have that $V_H^*(s_H) = \bigcup_{a \in \mathcal{A}} Q_H^*(s_H, a) = \bigcup_{a \in \mathcal{A}} \{\{\hat{r}(s_H, a)\}\} = \{\{\hat{r}(s_H, a)\} \mid a \in \mathcal{A}\}$. Thus, for any $\mathcal{R} \in V_H^*(s_H)$, there is a trajectory starting from $s_H$ whose multiset of discretized reward values is exactly $\mathcal{R}$. Suppose the claim is true for level $h + 1$. Consider level $h$ and a state $s \in \mathcal{S}$. Let $\mathcal{R} \in V_h^*(s_h)$. According to Algorithm 1, there exists $a \in \mathcal{A}$ such that $\mathcal{R} \in Q_h^*(s_h, a)$. Let $s_{h+1} = T(s_h, a)$. We know that $\mathcal{R} \setminus \{\hat{r}(s_h, a)\} \in V^*(s_{h+1})$. By the induction hypothesis, there is a trajectory starting from $s_{h+1}$ whose multiset of discretized reward values is exactly $\mathcal{R} \setminus \{\hat{r}(s_h, a)\}$. Thus, there exists a trajectory starting from $s_h$ whose multiset of discretized reward values is exactly $\mathcal{R}$.

Suppose there exists a trajectory starting from state $s$ and level $h$ whose multiset of discretized reward values is exactly $\mathcal{R}$. We want to show that $\mathcal{R} \in V_h^*(s)$. The proof is by induction on $h \in [H]$. Consider the base case when $h = H$. Let $s_H \in \mathcal{S}$. We have that $V_H^*(s_H) = \bigcup_{a \in \mathcal{A}} Q_H^*(s_H, a) = \bigcup_{a \in \mathcal{A}} \{\{\hat{r}(s_H, a)\}\} = \{\{\hat{r}(s_H, a)\} \mid a \in \mathcal{A}\}$. Thus, for any trajectory starting from $s_H$ whose multiset of discretized reward values is exactly $\mathcal{R}$, we have $\mathcal{R} \in V_H^*(s_H)$. Suppose the claim is true for level $h + 1$. Consider level $h$ and a state $s_h \in \mathcal{S}$ and a trajectory $s_h, a_h, \hat{r}_h, s_{h+1}, a_{h+1}, \hat{r}_{h+1}, \cdots, s_H, a_H, \hat{r}_H$ starting from $s_h$. Let $\mathcal{R} = \{\hat{r}_h, \hat{r}_{h+1}, \cdots, \hat{r}_H\}$. By induction hypothesis, we have $\mathcal{R} \setminus \{\hat{r}_h\} \in V_{h+1}^*(s_{h+1})$. According to Algorithm 1, we have $\mathcal{R} \in Q_h^*(s_h, a_h)$. Thus, we have $\mathcal{R} \in V_h^*(s_h)$. $\square$

**Claim A.3.** *Let $s_1, a_1, r_1, s_2, a_2, r_2, \cdots, s_H, a_H, r_H$ be the trajectory induced by policy $\pi_{\mathcal{R}}$. We have $\mathcal{R} = \{\hat{r}_1, \hat{r}_2, \cdots, \hat{r}_H\}$.*

*Proof.* We prove that for any $h \in [H]$, we have $\mathcal{R}^h \cup \{\hat{r}(s_i, \pi_{\mathcal{R}}(s_i)) \mid i \in [h-1]\} = \mathcal{R}$. The proof is by induction. For $h = 1$, it is true since $\mathcal{R}^1 = \mathcal{R}$. Furthermore, by Claim A.2, we know that there exists $a \in \mathcal{A}$ such that $\mathcal{R}^1 \in Q_1^*(s_1, a)$. Suppose it is true for $h - 1$. In the $h$-th iteration, we have $\mathcal{R}^{h-1} \in Q_h^*(s_h, \pi_{\mathcal{R}}(s_h))$ and $\mathcal{R}^h = \mathcal{R}^{h-1} \setminus \{\hat{r}(s_h, \pi_{\mathcal{R}}(s_h))\}$. Thus, we have $\mathcal{R} = \mathcal{R}^{h-1} \cup \{\hat{r}(s_i, \pi_{\mathcal{R}}(s_i)) \mid i \in [h-1]\} = \mathcal{R}^h \cup \{\hat{r}(s_i, \pi_{\mathcal{R}}(s_i)) \mid i \in [h-1]\} \cup \{\hat{r}(s_h, \pi_{\mathcal{R}}(s_h))\} = \mathcal{R}^h \cup \{\hat{r}(s_i, \pi_{\mathcal{R}}(s_i)) \mid i \in [h]\}$. Furthermore, by Claim A.2, we have $\mathcal{R}^h \in V_{h+1}^*(s_{h+1})$.

Notice that $\mathcal{R}^H = \emptyset$. Thus we can conclude the claim. $\qquad\square$

Now we formally prove Theorem A.1.

Let $s_1^*, a_1^*, r_1^*, s_2^*, a_2^*, r_2^*, \cdots, s_H^*, a_H^*, r_H^*$ be the trajectory induced by the optimal policy. Let $\mathcal{R}^* = \{r_h^* \mid h \in [H]\}$, and $\hat{\mathcal{R}} = \{\hat{r}_h \mid h \in [H]\}$ be the discretized version of $\mathcal{R}^*$. According to Claim A.2, we know that $\hat{\mathcal{R}} \in V^*(s_1)$. Let $\hat{s}_1, \hat{a}_1, \hat{r}_1, \hat{s}_2, \hat{a}_2, \hat{r}_2, \cdots, \hat{s}_H, \hat{a}_H, \hat{r}_H$ be the trajectory induced by the policy $\pi_{\hat{\mathcal{R}}}$ with discretized reward values. According to Claim A.3, we have $\hat{\mathcal{R}} = \{\hat{r}_1, \hat{r}_2, \cdots, \hat{r}_H\}$. Let $\hat{s}_1, \hat{a}_1, \tilde{r}_1, \hat{s}_2, \hat{a}_2, \tilde{r}_2, \cdots, \hat{s}_H, \hat{a}_H, \tilde{r}_H$ be the trajectory induced by the policy $\pi_{\hat{\mathcal{R}}}$ with original (undiscretized) reward values. Let $\tilde{\mathcal{R}} = \{\tilde{r}_1, \tilde{r}_2, \cdots, \tilde{r}_H\}$. By the choice of $\bar{\pi}$ outputted by Algorithm 1, we have:

$$
\begin{aligned}
f(\bar{\pi}) &\geq f(\pi_{\hat{\mathcal{R}}}) \\
&= f(\{\tilde{r}_1, \tilde{r}_2, \cdots, \tilde{r}_H\}) \\
&\geq f\left(\{\tilde{r}_h \cdot \mathbb{1}(\tilde{r}_h \geq \bar{\delta}) \mid h \in [H]\}\right) - \varepsilon/4 \\
&\geq f(\hat{\mathcal{R}}) - \varepsilon/2 \\
&\geq f\left(\{r_h^* \cdot \mathbb{1}(r_h^* \geq \bar{\delta}) \mid h \in [H]\}\right) - 3 \cdot \varepsilon/4 \\
&\geq f(\mathcal{R}^*) - \varepsilon,
\end{aligned}
$$

where the first step follows from $\hat{\mathcal{R}} \in V^*(s_1)$ and the choice of $\bar{\pi}$, the third step follows from that $f(\cdot)$ is $(\varepsilon/4, \bar{\delta})$-insensitive to small entries, the fourth step follows from that $f(\cdot)$ is $(\varepsilon/4, \hat{\delta})$-approximate homogeneous, the fifth step follows from that $f(\cdot)$ is $(\varepsilon/4, \hat{\delta})$-approximate homogeneous and the last step follows from that $f(\cdot)$ is $(\varepsilon/4, \bar{\delta})$-insensitive to small entries.

Thus, $\bar{\pi}$ is an $\varepsilon$-optimal policy. $\qquad\square$

# B    Lower Bounds

In this section, we formally prove our lower bounds. Let us first introduce a query problem called INDEX-QUERY which will be useful in our lower bound arguments.

**Definition B.1** (INDEX-QUERY, [16]). *In the* $\mathrm{INDQ}_n$ *problem, there is an underlying* $i^* \in [n]$. *The algorithm sequentially and adaptively outputs guesses* $i \in [n]$ *and queries whether* $i = i^*$. *The goal is to output* $i^*$, *using as few queries as possible.*

**Definition B.2** ($\delta$-correct algorithms, [16]). *For* $\delta \in (0, 1)$, *a randomized algorithm is* $\delta$-correct for $\mathrm{INDQ}_n$, *if for any underlying* $i^* \in [n]$, *with probability at least* $1 - \delta$, *the algorithm outputs* $i^*$.

The following theorem stats the query complexity of $\mathrm{INDQ}_n$ for 0.1-correct algorithms.

**Theorem B.3** (Theorem 5.1 of [16]). *Any* 0.1-*correct algorithm for* $\mathrm{INDQ}_n$ *requires at least* $0.9n$ *queries in the worst case.*

## B.1    Necessity of Symmetry

**Theorem B.4.** *There is a family* $\mathcal{F}$ *of objective functions which are* $(\varepsilon, \varepsilon)$-*approximate homogeneous and are* $(2\varepsilon, \varepsilon)$-*insensitive to small entires but are not necessarily symmetric, such that any algorithm which can output a* 0.49-*optimal policy for any objective function* $f \in \mathcal{F}$ *with probability at least* 0.9 *needs to query the objective values of at least* $0.9 \cdot 2^H$ *policies in the worst case.*

*Proof.* We describe our deterministic system as the following. For each $h \in [H]$, there is a single state $s_h \in \mathcal{S}_h$. There are two actions $a_1, a_2$ in the action space $\mathcal{A}$, and $T(s_h, a_1) = T(s_h, a_2) = s_{h+1}$ for any $1 \leq h < H$. The reward function satisfies that $r(s_h, a_1) = 1/2$ and $r(s_h, a_2) = 1$ for $h \in [H]$.

For a vector $\theta \in \mathbb{R}^H$, we define a function $f_\theta : [0, 1]^H \to [0, 1]^H$, if there exists $x_h = 0$ for some $h \in [H]$ then we define $f_\theta(x) = 0$. Otherwise,

$$
f_\theta(x) = \min_{h \in [H]} \min(x_h/\theta_h, \theta_h/x_h).
$$

Let $\mathcal{F} = \{f_\theta \mid \theta \in \{1/2, 1\}^H\}$. Firstly, we show that for any $f \in \mathcal{F}$, $f$ is $(\varepsilon, \varepsilon)$-approximate homogeneous for any $\varepsilon \geq 0$. Let $\varepsilon \geq 0$. Consider two vectors $x, y \in [0,1]^H$ such that for any $h \in [H]$, $x_h \in [y_h, (1+\varepsilon)y_h]$. For $h \in [H]$, if $x_h \leq \theta_h$, then $y_h/\theta_h \leq x_h/\theta_h$, $y_h/\theta_h \geq x_h/\theta_h/(1+\varepsilon) \geq (1-\varepsilon) \cdot (x_h/\theta_h) \geq x_h/\theta_h - \varepsilon$, and $\theta_h/y_h \geq \theta_h/x_h \geq 1$. If $x_h \geq \theta_h$, then $\theta_h/y_h \geq \theta_h/x_h$, $\theta_h/y_h \leq (1+\varepsilon) \cdot \theta_h/x_h \leq \theta_h/x_h + \varepsilon$, and $y_h/\theta_h \geq 1/(1+\varepsilon) \geq 1 - \varepsilon$. Thus, $f_\theta(y) \in [f_\theta(x) - \varepsilon, f_\theta(x) + \varepsilon]$. Next, we show that for any $f \in \mathcal{F}$, $f$ is $(2\varepsilon, \varepsilon)$-insensitive to small entries. Let $\varepsilon \geq 0$. Consider any $f_\theta \in \mathcal{F}$, and any $x \in [0,1]^H$, $h \in [H]$ with $x_h \leq \varepsilon$. If $\theta_h = 1$, then $\min(x_h/\theta_h, \theta_h/x_h) \leq \varepsilon$. If $\theta_h = 1/2$, then $\min(x_h/\theta_h, \theta_h/x_h) \leq 2\varepsilon$. Thus, $f_\theta$ is $(2\varepsilon, \varepsilon)$-insensitive to small entries.

Consider an arbitrary vector $x \in \{1/2, 1\}^H$ and a function $f_\theta \in \mathcal{F}$. If $x = \theta$, then by the definition of $f_\theta$, we know that $f_\theta(x) = 1$. If $x \neq \theta$, let us consider any $h \in [H]$ such that $x_h \neq \theta_h$. If $x_h = 1/2$ and $\theta_h = 1$, then $\min(x_h/\theta_h, \theta_h/x_h) = 1/2$. If $x_h = 1$ and $\theta_h = 1/2$, then $\min(x_h/\theta_h, \theta_h/x_h) = 1/2$. Thus, $f_\theta(x) = 1/2$ when $x \neq \theta$.

Now consider a policy $\pi$ and an objective function $f_\theta \in \mathcal{F}$. Let $s_1, a_1, r_1, s_2, a_2, r_2, \cdots, s_H, a_H, r_H$ be the trajectory induced by $\pi$. Let $x = (r_1, r_2, \cdots, r_H)$. We know that the optimal policy for $f_\theta$ is the policy with $x = \theta$, and in that case we have $f_\theta(\pi) = 1$. For any non-optimal policy $\pi$ we know that $f_\theta(\pi) = 1/2$.

Now it suffices to prove the theorem. Our proof is by reduction from $\mathrm{INDQ}_{2^H}$. Suppose we have an algorithm $\mathcal{M}$ which outputs $0.49$-optimal policy for any $f \in \mathcal{F}$. We will show that there is a query algorithm for $\mathrm{INDQ}_{2^H}$. In problem $\mathrm{INDQ}_{2^H}$, there is an underlying $\theta^* \in \{1/2, 1\}^H$ and we want to find $\theta^*$. We can imagine that the deterministic system has objective function $f_{\theta^*} \in \mathcal{F}$ and then we simulate $\mathcal{M}$. Suppose the $i$-th query policy of $\mathcal{M}$ is $\pi$, then we let $x = (r_1, r_2, \cdots, r_H)$ be the reward values induced by $\pi$. Then we query whether $x = \theta^*$. If the answer is yes, then we are done. Otherwise, since $f_{\theta^*}(x) = 1/2$ for $x \neq \theta^*$, we can return an objective value of $1/2$ for the $i$-th query of $\mathcal{M}$ and continue the simulation of $\mathcal{M}$. Since $\mathcal{M}$ can output a $0.49$-optimal policy with probability at least $0.9$, it must output the optimal policy with probability at least $0.9$ which means that it can eventually find $x = \theta^*$ with probability at least $0.9$. According to Theorem B.3, $\mathcal{M}$ must query at least $0.9 \cdot 2^H$ policies for the worst $f \in \mathcal{F}$. $\qquad\square$

**Theorem B.5.** *There is a family $\mathcal{F}$ of objective functions which are symmetric and are $(0, \varepsilon)$-insensitive to small entires for any $\varepsilon \leq 1/2$ but are not necessarily approximate homogeneous, such that any algorithm which can output a $0.99$-optimal policy for any objective function $f \in \mathcal{F}$ with probability at least $0.9$ needs to query the objective values of at least $0.9 \cdot 2^H$ policies in the worst case.*

*Proof.* We describe our deterministic system as the following. For each $h \in [H]$, there is a single state $s_h \in \mathcal{S}_h$. There are two actions $a_1, a_2$ in the action space $\mathcal{A}$, and $T(s_h, a_1) = T(s_h, a_2) = s_{h+1}$ for any $1 \leq h < H$. The reward function satisfies that $r(s_h, a_1) = (2H + 2h - 1)/4H$ and $r(s_h, a_2) = (2H + 2h)/4H$ for $h \in [H]$.

Now we define $f_\theta$, which is parameterized by a vector $\theta \in \mathbb{R}^H$. For any vector $x \in [0,1]^H$, we use $i_1, i_2, \ldots, i_H$ to denote a permutation of $(1, 2, \ldots, H)$ such that $0 \leq x_{i_1} \leq x_{i_2} \leq \ldots \leq x_{i_H} \leq 1$. We define $f_\theta(x) = 1$ if $x_{i_h} = \theta_h$ for all $h \in [H]$, and $f_\theta(x) = 0$ otherwise. Let

$$\mathcal{F} = \left\{ f_\theta \mid \theta \in \mathbb{R}^H, \forall h \in [H], \theta_h \in \{(2h + 2H - 1)/4H, (h + H)/2H\} \right\}.$$

By construction, $f_\theta$ is clearly symmetric. Since for any $f_\theta \in \mathcal{F}$, each entry of $\theta$ is greater than $1/2$, $f_\theta(x)$ must be $0$ if $x$ has any entry at most $1/2$ which implies that $f_\theta$ is $(0, \varepsilon)$-insensitive to small entries for any $\varepsilon \leq 1/2$.

Consider an arbitrary vector $x \in \mathbb{R}^H$ and a function $f_\theta \in \mathcal{F}$. Without loss of generality, we can assume $x_1 \leq x_2 \leq \cdots \leq x_H$. If $x = \theta$, then $f_\theta(x) = 1$. Otherwise, $f_\theta(x) = 0$ according to the definition of $f_\theta$.

Now consider a policy $\pi$ and an objective function $f_\theta \in \mathcal{F}$. Let $s_1, a_1, r_1, s_2, a_2, r_2, \cdots, s_H, a_H, r_H$ be the trajectory induced by $\pi$. Let $x = (r_1, r_2, \cdots, r_H)$. We know that the optimal policy for $f_\theta$ is the policy with $x = \theta$. For any non-optimal policy $\pi$ we know that $f_\theta(\pi) = 0$.

Now it suffices to prove the theorem. Our proof is by reduction from $\mathrm{INDQ}_{2^H}$. Suppose we have an algorithm $\mathcal{M}$ which outputs $0.99$-optimal policy for any $f \in \mathcal{F}$. We will show that there is a query algorithm for $\mathrm{INDQ}_{2^H}$. In problem $\mathrm{INDQ}_{2^H}$, there is an underlying $\theta^* \in \mathbb{R}^H$ satisfying for any

$h \in [H], \theta_h \in \{(2H + 2h - 1)/4H, (H + h)/2H\}$ and we want to find $\theta^*$. We can imagine that the deterministic system has objective function $f_{\theta^*} \in \mathcal{F}$ and then we simulate $\mathcal{M}$. Suppose the $i$-th query policy of $\mathcal{M}$ is $\pi$, then we let $x = (r_1, r_2, \cdots, r_H)$ be the reward values induced by $\pi$. Due to the construction of our deterministic system, we have $r_1 \le r_2 \le \cdots \le r_H$. Then we query whether $x = \theta^*$. If the answer is yes, then we are done. Otherwise, since $f_{\theta^*}(x) = 0$ for $x \ne \theta^*$, we can return an objective value of $0$ for the $i$-th query of $\mathcal{M}$ and continue the simulation of $\mathcal{M}$. Since $\mathcal{M}$ can output a 0.99-optimal policy with probability at least 0.9, it must output the optimal policy with probability at least 0.9 which means that it can eventually find $x = \theta^*$ with probability at least 0.9. According to Theorem B.3, $\mathcal{M}$ must query at least $0.9 \cdot 2^H$ policies for the worst $f \in \mathcal{F}$. $\qquad \square$

**Theorem B.6.** *There is a family $\mathcal{F}$ of objective functions which are symmetric and are $(\varepsilon, \varepsilon)$-approximate homogeneous for any $\varepsilon \ge 0$ but are not necessarily insensitive to small entries, such that any algorithm which can output a 0.49-optimal policy for any objective function $f \in \mathcal{F}$ with probability at least 0.9 needs to query the objective values of at least $0.9 \cdot 2^H$ policies in the worst case.*

*Proof.* We describe our deterministic system as the following. For each $h \in [H]$, there is a single state $s_h \in \mathcal{S}_h$. There are two actions $a_1, a_2$ in the action space $\mathcal{A}$, and $T(s_h, a_1) = T(s_h, a_2) = s_{h+1}$ for any $1 \le h < H$. The reward function satisfies that $r(s_h, a_1) = 2^{-H(2h-1)}$ and $r(s_h, a_2) = 2^{-2Hh}$ for $h \in [H]$.

Now we define $f_\theta$, which is parameterized by a vector $\theta \in \mathbb{R}^H$. For any vector $x \in [0, 1]^H$, we use $i_1, i_2, \ldots, i_H$ to denote a permutation of $(1, 2, \ldots, H)$ such that $1 \ge x_{i_1} \ge x_{i_2} \ge \ldots \ge x_{i_H} \ge 0$. We define

$$f_\theta(x) = \min_{h \in [H]} \min(x_{i_h}/\theta_h, \theta_h/x_{i_h}).$$

Let

$$\mathcal{F} = \left\{ f_\theta \mid \theta \in \mathbb{R}^H, \forall h \in [H], \theta_h \in \{2^{-H(2h-1)}, 2^{-2Hh}\} \right\}.$$

By construction, $f_\theta$ is symmetric. Next, we show that for any $f \in \mathcal{F}$, $f$ is $(\varepsilon, \varepsilon)$-approximate homogeneous for any $\varepsilon \ge 0$. Consider two vectors $x, y \in [0, 1]^H$ such that for any $h \in [H], x_h \in [y_h, (1 + \varepsilon)y_h]$. We use $i_1, i_2, \ldots, i_H$ to denote a permutation of $(1, 2, \ldots, H)$ such that $1 \ge x_{i_1} \ge x_{i_2} \ge \ldots \ge x_{i_H} \ge 0$. We use $i'_1, i'_2, \ldots, i'_H$ to denote a permutation of $(1, 2, \ldots, H)$ such that $1 \ge y_{i'_1} \ge y_{i'_2} \ge \ldots \ge y_{i'_H} \ge 0$. We claim that for any $h \in [H]$, we have $x_{i_h} \in [y_{i'_h}, (1+\varepsilon)y_{i'_h}]$. The reason is as follows. Because $x_{i'_1} \ge y_{i'_1} \ge y_{i'_h}, x_{i'_2} \ge y_{i'_2} \ge y_{i'_h}, \cdots, x_{i'_h} \ge y_{i'_h}$, the $h$-th largest value $x_{i_h}$ in $x_1, x_2, \cdots, x_H$ must be at least $y_{i'_h}$. Because $x_{i'_H}/(1 + \varepsilon) \le y_{i'_H} \le y_{i'_h}, x_{i'_{H-1}}/(1 + \varepsilon) \le y_{i'_{H-1}} \le y_{i'_h}, \cdots, x_{i'_h}/(1 + \varepsilon) \le y_{i'_h}$, the $(H - h + 1)$-th smallest value $x_{i_h}$ in $x_1, x_2, \cdots, x_H$ must be at most $(1 + \varepsilon)y_{i'_h}$. For $h \in [H]$, if $x_{i_h} \le \theta_h$, then $y_{i'_h}/\theta_h \le x_{i_h}/\theta_h$, $y_{i'_h}/\theta_h \ge x_{i_h}/\theta_h/(1+\varepsilon) \ge (1-\varepsilon) \cdot (x_{i_h}/\theta_h) \ge x_{i_h}/\theta_h - \varepsilon$, and $\theta_h/y_{i'_h} \ge \theta_h/x_{i_h} \ge 1$. If $x_{i_h} \ge \theta_h$, then $\theta_h/y_{i'_h} \ge \theta_h/x_{i_h}, \theta_h/y_{i'_h} \le (1 + \varepsilon) \cdot \theta_h/x_{i_h} \le \theta_h/x_{i_h} + \varepsilon$, and $y_{i'_h}/\theta_h \ge 1/(1 + \varepsilon) \ge 1 - \varepsilon$. Thus, $f_\theta(y) \in [f_\theta(x) - \varepsilon, f_\theta(x) + \varepsilon]$.

Consider an arbitrary vector $x \in \mathbb{R}^H$ satisfying for any $h \in [H], x_h \in \{2^{-H(2h-1)}, 2^{-2Hh}\}$ and a function $f_\theta \in \mathcal{F}$. It is easy to see that we always have $1 \ge x_1 \ge x_2 \ge \cdots \ge x_H \ge 0$. If $x = \theta$, then $f_\theta(x) = 1$. Otherwise, consider any $h \in [H]$ such that $x_h \ne \theta_h$. If $x_h = 2^{-H(2h-1)}, \theta_h = 2^{-2Hh}$, then $\min(x_h/\theta_h, \theta_h/x_h) = 2^{-H}$. If $\theta_h = 2^{-H(2h-1)}, x_h = 2^{-2Hh}$, then $\min(x_h/\theta_h, \theta_h/x_h) = 2^{-H}$. Thus, $f_\theta(x) = 2^{-H}$ if $x \ne \theta$.

Now consider a policy $\pi$ and an objective function $f_\theta \in \mathcal{F}$. Let $s_1, a_1, r_1, s_2, a_2, r_2, \cdots, s_H, a_H, r_H$ be the trajectory induced by $\pi$. Let $x = (r_1, r_2, \cdots, r_H)$. We know that the optimal policy for $f_\theta$ is the policy with $x = \theta$. For any non-optimal policy $\pi$ we know that $f_\theta(\pi) = 2^{-H}$.

Now it suffices to prove the theorem. Our proof is by reduction from $\text{INDQ}_{2^H}$. Suppose we have an algorithm $\mathcal{M}$ which outputs 0.49-optimal policy for any $f \in \mathcal{F}$. We will show that there is a query algorithm for $\text{INDQ}_{2^H}$. In problem $\text{INDQ}_{2^H}$, there is an underlying $\theta^* \in \mathbb{R}^H$ satisfying for any $h \in [H], \theta_h \in \{2^{-H(2h-1)}, 2^{-2Hh}\}$ and we want to find $\theta^*$. We can imagine that the deterministic system has objective function $f_{\theta^*} \in \mathcal{F}$ and then we simulate $\mathcal{M}$. Suppose the $i$-th query policy of $\mathcal{M}$ is $\pi$, then we let $x = (r_1, r_2, \cdots, r_H)$ be the reward values induced by $\pi$. By our construction of the

deterministic system we have $1 \geq r_1 \geq r_2 \geq \cdots \geq r_H \geq 0$. Then we query whether $x = \theta^*$. If the answer is yes, then we are done. Otherwise, since $f_{\theta^*}(x) = 2^{-H}$ for $x \neq \theta^*$, we return an objective value of $2^{-H}$ for the $i$-th query of $\mathcal{M}$ and continue the simulation of $\mathcal{M}$. Since $\mathcal{M}$ can output a $0.49$-optimal policy with probability at least $0.9$, it must output the optimal policy with probability at least $0.9$ which means that it can eventually find $x = \theta^*$ with probability at least $0.9$. According to Theorem B.3, $\mathcal{M}$ must query at least $0.9 \cdot 2^H$ policies for the worst $f \in \mathcal{F}$. $\qquad\square$