[Reviews · NeurIPS 2020]

Review 1

Summary and Contributions: This paper addresses planning in deterministic Markov decision processes with alternative objective functions (as opposed to the standard expected value). The authors give three conditions required for the objective function, under which their algorithm can be used: symmetry, approximate homogeneity, and insensitivity to small entries. The first condition means that the order that rewards are received doesn't matter in the objective function. The other conditions allow the rewards to be discretised while only inducing a small error in the objective function. The authors present two algorithms for specific cases, and a general algorithm for general objective functions which adhere to these criteria. The general algorithm involves discretising the reward function, and then enumerating every possible set of rewards which could have been received over the horizon. Then the policy is extracted which corresponds to the set of rewards which optimises the particular objective function chosen.

Strengths: The paper is clearly written and covers quite a relevant topic. The authors elegantly describe the problem and are able to show that all the required conditions over the general reward function are required. Furthermore, these conditions still cover a number of objective functions which are relevant to the planning community. I believe that the algorithms presented will correctly optimise the objective functions within the approximation errors given.

Weaknesses: The paper only addresses deterministic MDPs, which drastically limits its usefulness, and no insight is given on how it can be relaxed. Section 5 - algorithm for general symmetric norms. The first few sentences of the second paragraph of "High level ideas" was difficult to understand. It took me a while to realise that the idea is to compute policies which each only optimise for one of the discretised reward values, and then choose the best policies of these. This seems to be a very naive algorithm, which is evidenced by the wide error bound provided. The algorithm in Section 6 is also rather brute force, requiring full enumeration of all of the possible sets of rewards accumulated. There are no experiments to showcase the approach. Whilst I believe them to be relevant, the motivating example for the need for general reward functions is weak.

Correctness: I am concerned with the correctness of the complexity result. The multisets of discretised reward values contain at most H rewards, each of which takes upon one of log(1/delta1)/delta2 values. The authors say that the number of possible multisets is upper bounded by H^(log(1/delta1)/delta2), thus giving a complexity polynomial in H. However, my impression is that the number of multisets should be upper bounded by (log(1/delta1)/delta2)^H. This correctness of this result is highly important to the contribution of the paper, as the main claim is that the algorithm is polynomial in H. Could the authors please explain how they get the upper bound of H^(log(1/delta1)/delta2). Unfortunately there is no empirical evaluation.

Clarity: I thought the paper was moderately well written overall. Specifically, I thought section 1 was very well written, particularly the contributions section. I found the sections describing the algorithms more difficult to understand. I think Section 5 would have been easier to understand if it had used pseudocode, or some kind of more formal description as opposed to the wordy description given.

Relation to Prior Work: There are other approaches that consider different optimisation objectives for MDPs. Examples are constrained MDPs, and RL/planning with temporal logic (non-Markovian) goals and constraints. These should be discussed.

Reproducibility: Yes

Additional Feedback: Post-rebuttal: I was happy about the author's response regarding the complexity result. Also, whilst I still think that in practice it will be hard to scale the proposed algorithms to realistic applications and I'm not a fan of the deterministic MDP assumption, I do appreciate the general setup of the paper. Given that, the other reviews, and the author's response, I increased my score. I'm not convinced from the theoretical results that the algorithm presented can actually practically be implemented. Assuming that the complexity result is correct, the objective function must be evaluated up to H^(log(1/delta1)/delta2) times by the algorithm, which, despite being a polynomial of H, is still a very large number of evaluations. For example, setting a planning horizon of 20 and both deltas to 0.02 would mean that the number of function evaluations required is up to 20^85. Because of this, I think the paper would have been stronger if it did include experimental results showing the scalability of the algorithm. I think the paper would have been better if Sections 4 and 5 were removed (as I did not think these sections contributed much to the paper as noted above) and instead, experimental results of the main algorithm were included.


Review 2

Summary and Contributions: The paper is about summarizing single-step rewards over a finite horizon in a deterministic environment. Instead of focusing on summing rewards, as the vast majority of work in the area does, it considers a much more general class of summaries. It provides an approximation algorithm for summary functions that satisfy particular technical properties, good intuitions for how this algorithm works, and an argument that the technical properties are necessary to achieve sub-exponential learning/planning.

Strengths: The setting is interesting and the results informative and insightful.

Weaknesses: The idea of generalizing away from sums of rewards seems a little far-fetched to me, but it's intriguing. It would have been nice if the resulting algorithm generalized value iteration in a way that resulted in recovering value iteration when the summary function is the sum. More significantly, I didn't like that there was no discussion about how the framework might be made to include stochastic transitions, which are considered pretty central to the study of MDPs. I'd be interested in some reflection on what makes the stochastic case different---what prevents these techniques from being applied in environments with stochastic transitions/rewards?

Correctness: Yes, the results seem solid.

Clarity: It's sufficiently clear.

Relation to Prior Work: Good connections to related work are included.

Reproducibility: Yes

Additional Feedback: Post feedback response: I appreciate the author feedback. One item I want to flag, though. The feedback said (of one of the reviews): "We are grateful to the reviewer for providing a comprehensive list of papers on non-Markovian reward". I do not think the list is at all "comprehensive". It represents a number of very relevant and very significant papers, but there are others in this area. Thanks! Detailed comments: "is a much robust objective" -> "is a much more robust objective"? "states that for any input" -> "states that, for any input" "two large family" -> "two large families" "to such objective function" -> "to such an objective function" "polynomial-time approximation scheme": Include a reference? "sample complexity and regret bound" -> "sample complexity and regret bounds"? "line of research consider" -> "line of research considers" "a fundamental objective function": I'm not sure what "fundamental" means here. Reword? "can not simply use" -> "cannot simply use" "since the objective function depends on the whole trajectory.": Confusing. Surely, you can also say the sum of rewards depends on the whole trajectory, too, right? Maybe make the argument a different way? "a dynamic programming algorithm with augmented state space" -> "a dynamic programming algorithm with an augmented state space"? "Similar to standard dynamic programming approach for planning" -> "Similar to standard dynamic programming approaches for planning"? "that after the discretization, there" -> "that, after the discretization, there" "all rewards values" -> "all reward values" (many times) "dynamic programming with augmented state space" -> "dynamic programming with an augmented state space" "f is (eps/4, delhat)-insensitive to small entries and satisfy" -> "f is (eps/4, delhat)-insensitive to small entries and satisfies" "that without any of the three assumptions," -> "that, without any of the three assumptions," "exponential many" -> "exponentially many" "For a vector theta satisfies the above condition" -> "For a vector theta satisfying the above condition" "we study planning problems" -> "we studied planning problems"? References: Double check capitalization (for example, "rl agent using ltl")


Review 3

Summary and Contributions: This paper considers planning problems, in particular to optimize a certain class of objective functions in a Markov decision process (MDP). The main contribution of this paper is to generalize the typical problem of maximizing the expected reward over a finite planning horizon. As an example, the author's framework is (opposed to standard approaches) able to maximize the *minimal* reward, or to determine the k-th largest value among the rewards collected along the trajectory. The authors define a general framework for such measures and provide three necessary conditions for finite-state MDPs with a deterministic transition function. They also provide an algorithm that is linear in the number of states and actions in the MDP, but it is exponential in the horizon length. The main idea of the proposed algorithm is to discretize reward values into a number of intervals, as the algorithm scales exponentially in the number of different reward values. They also show the hardness results if any of the necessary conditions are violated.

Strengths: This paper provides three easy to check conditions that the algorithm requires, which provides the theoretical basis of the work. They also provide insights on applying non-uniform discretization and augmenting state space techniques to provide algorithms with guarantees.

Weaknesses: The main limitations are (1) the paper lacks an empirical evaluation or a good motivation, (2) the paper assumes deterministic transition functions, (3) the approach considers finite-horizon problems, and does not comment on contraction mappings. The latter is arguably the least important drawback, although recall that the method scales exponentially with the horizon length. In more detail, the size of all possible value functions induced by all policies increases exponentially in the planning horizon, which may limit the applicability of the algorithm. Additionally, the size may increase significantly if the transition function is non-deterministic, which further limits the applicability of the algorithm for most problems of interest. Most importantly, I am missing (1) a better motivation. While I believe that the work concerns interesting problems, it would be nice to see a well elaborated case study, and (even better) an implementation which indicates the practical applicability.

Correctness: I checked the steps of Theorem A.1. in the supplementary material, which is the main contribution of the paper, and I can follow the steps, which relies on the assumptions to provide the bounds. However, I was unable to check the proofs of symmetry that are in the supplementary material, as the proofs were very hard to follow, and the authors do not provide the steps of the proof for better readability.

Clarity: I think the authors can significantly improve the presentation, particularly explain the high-level ideas in steps provided in Section 4 and 5, and state the main ideas in the beginning of the section, not in the end. For example, I was very confused how the authors augment the state space in Section 4, and why the authors discretize the reward value uniformly, which was not the case before. Additionally, the presentations of the proofs in the supplementary material can significantly be improved, as some paragraphs in the proofs involve several steps without explaining them.

Relation to Prior Work: The contributions are sound, however, the authors should discuss why these objectives cannot be modeled by some of the mentioned approaches, such as using temporal logic, reward machines, or risk-sensitive measures. However, the study of a (more) general objective function is novel compared to existing work.

Reproducibility: No

Additional Feedback: After reading the author response, I think the paper can be accepted. The authors should make sure to add all promised content to the final version of the paper.


Review 4

Summary and Contributions: The authors define a family of non-Markovian reward functions for the situation where a single agent acts for a fixed finite number of steps, and then receives a reward. This family of reward functions is studied in the setting of deterministic state transition models that are given according to an explicit, tabular, representation -- i.e. all states enumerated. The family of functions the authors define satisfy 3 properties: (Definition 1.1) the agent's reward is determined according to the _set_ of states encountered in a finite trace -- i.e. the agents reward is invariant to the order in which states are encountered. The family of functions described are evaluated over a sequence of numerical state features. The authors' family of reward functions is insensitive to small numerical values, and satisfies an "approximate homogeneity" property - i.e. the range of the function is bound (additive) by \epsilon as we (multiplicatively) scale the numerical inputs, independently, within a given band. Those items are covered by Definitions 1.2 and 1.3. Treating their non-Markovian family of reward functions, the authors develop a PTAS (i.e. tractable and approximately optimal) dynamic programming algorithm for scenarios where the underlying system dynamics are characterised by a tabular representation of a finite deterministic state transition model. On their way to doing this, they treat two specific reward functions from their family. The associated analytical runtime analysis is provided in all cases. The exhibition of these preliminary examples is quite informative, and gives as the gist of what is to come. The authors show that each of their 3 reward function axioms are required for a PTAS algorithm. If you completely omit any one of the axioms, then no general PTAS algorithm exists for the resulting family. The language around this claim could be improved in the paper, but overall it is clear what is being claimed.

Strengths: The claims are important, and appear solid to me. There are some issues with consideration of related literature, but these do not diminish the authors theoretical claims, which are sufficiently interesting for a conference paper in my view.

Weaknesses: The study of the relationship to past works treating non-Markovian rewards is weak. More on this below. The work would benefit from some more compelling motivations from applications and industry. There could be some improvements to the presentation, are care around the use of language. More on this below.

Correctness: They seem OK to me. There is no empirical work here.

Clarity: Yes, generally OK. Abstract: "technical conditions" -- I would suggest --> "axiomatic conditions". L54: it is not clear that there is no "loss of generality". Constraining numeric features on states to be positive seems to be lossy to me. [THANKS AUTHORS FOR CLEARING THIS UP, I LOOK FORWARD TO READING FINAL MANUSCRIPT...] "query": Authors use the term "query" a number of times. For example, in Section 7 -- "then _any_ algorithm still needs to query". Here, it is clear the authors are writing about PTAS algorithms, however it is entirely unclear what the intuition for query is, and why this assists in the exposition of your proof strategy. So, some text around intuitions here, perhaps even following the proof sketch, would be nice... L309: "exponential number of values". Some more careful language here would be appreciated. You can speak to the cardinality of the range of f. From my understating, that's the cue you are seeking wrt the intuition you are expressing. L317: "all rewards values" --> "all reward values" Conclusions: Authors claim a number of times that their characterisation of objective functions is "general". But authors also acknowledge that they are treating a specific family---defined according to their 3 axioms---of non-Markovian reward functions. For the sake of keeping claims tight, and for the record, I would strongly encourage authors to use more precise language around your claims. Broader Impact: I can see clear value in the research direction here. I find the self-driving car motivational example to be a bit contrived. Generally, I think readers would appreciate something more concrete and realistic. For example, a scenario that the authors are proactively studying as part of an engagement with an industrial partner. Broader Impact: A note about "safety" guarantees. "Safety" refers to a class of system properties studied in the model checking (and sometimes AI Planning) literature called.. "safety" properties. The authors characterisation of safety is quite different, and they might consider adopting a different term of avoid confusion.

Relation to Prior Work: Related Work: Here, the treatment of the non-Markovian reward literature is too limited, and discussion on this topic misleading when you consider all the literature. The claims made are arguably correct wrt the literature considered by the authors, but false in the context of the wider literature. Especially given that many of the reward functions contemplated here could be coded straightforwardly in one of the two linear temporal logics, PLTL or $FLTL, and given that dynamic programming schemes have been described for those settings, some more care and rigour here wrt related work is required. [NOTED, THE REWARD APPROXIMATION CONSIDERED IN THIS WORK HAS NOT BEEN CONTEMPLATED IN THE BELOW LITERATURE] The early schemes are summarised and studied in [1]. Consideration of minimisation of extended-state automaton is studied in [2], and in earlier works in this literature that employ a past looking linear temporal logic to describe non-Markovian rewards. An extension of this literature to non-Markovian rewards that are characterised by regular languages is given in [3]. In cases where you can express the non-Markovian rewards in the submitted manuscript using the formalism in the LTL literature I have just cited, it would be interesting to see the solution algorithms compared, theoretically, or otherwise. For example, reward functions for kth biggest value (encountered), min value, and max value, can all be described using the linear temporal logic formalisms here, and correspond to reward schemes characterised by non-counting regular languages. This seminal literature on the topic of non-Markovian rewards should not be omitted in the discussion of related work. [1] @article{DBLP:journals/jair/ThiebauxGSPK06, author = {Sylvie Thi{\'{e}}baux and Charles Gretton and John K. Slaney and David Price and Froduald Kabanza}, title = {Decision-Theoretic Planning with non-Markovian Rewards}, journal = {J. Artif. Intell. Res.}, volume = {25}, pages = {17--74}, year = {2006}, url = {https://doi.org/10.1613/jair.1676}, doi = {10.1613/jair.1676}, timestamp = {Wed, 25 Sep 2019 18:01:09 +0200}, biburl = {https://dblp.org/rec/journals/jair/ThiebauxGSPK06.bib}, bibsource = {dblp computer science bibliography, https://dblp.org} } [2] @article{DBLP:journals/igpl/Slaney05, author = {John K. Slaney}, title = {Semipositive {LTL} with an Uninterpreted Past Operator}, journal = {Log. J. {IGPL}}, volume = {13}, number = {2}, pages = {211--229}, year = {2005}, url = {https://doi.org/10.1093/jigpal/jzi015}, doi = {10.1093/jigpal/jzi015}, timestamp = {Fri, 06 Mar 2020 22:00:04 +0100}, biburl = {https://dblp.org/rec/journals/igpl/Slaney05.bib}, bibsource = {dblp computer science bibliography, https://dblp.org} } [3] @inproceedings{DBLP:conf/pricai/Gretton14, author = {Charles Gretton}, title = {A More Expressive Behavioral Logic for Decision-Theoretic Planning}, booktitle = {{PRICAI} 2014: Trends in Artificial Intelligence - 13th Pacific Rim International Conference on Artificial Intelligence, Gold Coast, QLD, Australia, December 1-5, 2014. Proceedings}, pages = {13--25}, year = {2014}, crossref = {DBLP:conf/pricai/2014}, url = {https://doi.org/10.1007/978-3-319-13560-1\_2}, doi = {10.1007/978-3-319-13560-1\_2}, timestamp = {Tue, 14 May 2019 10:00:46 +0200}, biburl = {https://dblp.org/rec/conf/pricai/Gretton14.bib}, bibsource = {dblp computer science bibliography, https://dblp.org} }

Reproducibility: Yes

Additional Feedback: It is weird to talk about "reward" as the numeric feature of a state here, and "reward" as the agent's reward at the end of a trace. I suggest changing the language [TO TALK ABOUT NUMERIC STATE FEATURES].

[Author Response · NeurIPS 2020]

We thank all the reviewers for their valuable feedback. We first address some common concerns.

**Deterministic systems.** Although deterministic systems seem restrictive in theory, in practice, lots of RL problems are
indeed deterministic. Also, this assumption makes the problem more tractable. We will emphasize that we focus on
deterministic systems in the next version, and will also leave extending our result to stochastic environments as an open
problem.

**Practicality.** We stress that our goal is to design provably efficient algorithms for RL with general reward functions.
In this paper, we focus on giving sufficient and necessary conditions that admit efficient algorithms, and we believe
our algorithmic insights (discretization, augmenting state space) can be applied in practice, which we are currently
exploring. However, this is not the focus of the current work.

—— **To Reviewer #1** ——

**Section 5.** The main goal of the algorithm in Section 5 is to motivate the discretization procedure used in the more
complicated algorithm in Section 6, and thus we do not focus on optimizing the approximation guarantee. In the next
version we will provide pseudocode in Section 5 to make the description of the algorithm formal.

**Correctness of the complexity result / The algorithm in Section 6 is also rather brute force.** The number of
multisets of cardinality $k$, with elements taken from a finite set of cardinality $n$, is $\binom{n+k-1}{k} \leq (k+1)^n$. This bound
can be found on the wikipedia page of multiset. For our case, $k = H$ and $n = \Theta(\log(1/\hat{\delta})/\bar{\delta})$, and thus the number of
possible multisets is at most $H^{\Theta(\log(1/\hat{\delta})/\bar{\delta})}$. Notice that "multisets" are different from "sequences", i.e., for multisets
we do not care about orders of elements. Indeed, the number of sequences of length $k$, with elements taken from a
finite set of cardinality $n$, could be as large as $n^k$. This also explains why our algorithm is not brute force: we carefully
discretize reward values to make the number of possible elements small, and we exploit the symmetric of the objective
function so that we only need to deal with multisets instead of sequences to avoid an exponential dependency on $H$.

—— **To Reviewer #2** ——

We are grateful to the reviewer for providing detailed comments on the writing of our paper, and we will revise the
paper according to the reviewer's comment in the next version.

**What prevents these techniques from being applied in stochastic environments?** Consider the following case in
the symmetric norm setting (Section 5 in our paper), which suggests that the stochastic case is fundamentally more
difficulty: there are two actions at the initial state, and all further actions do not affect the rewards. If the first action is
chosen, then half of the reward values will be 1 and half of the reward values will be 0. If the second action is chosen,
then all reward values will be a fair coin (0 or 1 with equal probability). To find a near-optimal policy, the agent must
carefully compare the expected objective values of both choices and thus cannot be handled by our algorithm. However,
if rewards are deterministic, one can simply return the action with more reward values of 1 which is optimal. Thus,
even for the setting that rewards are stochastic and transitions are deterministic, the problem becomes much harder.

—— **To Reviewer #3** ——

**Finite-horizon problems / exponential in the horizon length.** We would like to remind the reviewer that the running
time of our algorithm is polynomial in the planning horizon $H$ instead of being exponential in $H$. See Theorem 4.1 for
the precise statement. One can reduce discounted MDPs to finite-horizon MDPs by considering the first $\widetilde{O}(1/(1-\gamma))$
levels, and the $H$ dependency in the complexity of our algorithm will be replaced by $\widetilde{O}(1/(1-\gamma))$.

**Improve the presentation.** We will revise the paper, make the proofs more readable and improve the presentation in
general according to the reviewer's comments in the next version. In particular, we will explain the high-level ideas at
the beginning of Section 4 and 5 instead of at the end.

—— **To Reviewer #4** ——

**Improvements to the presentation / care around the use of language / Broader Impact.** We are grateful to the
reviewer for providing detailed comments on the writing of our paper, and we will revise the paper according to the
reviewer's comments in the next version.

**"query" / "exponential number of values".** We will make it clear in the next version that we are proving lower bounds
on the number of times that the agent evaluates the objective function $f$. Here by a "query", we mean that the agent
evaluates the objective function $f$ on some specific input.

**Literature on non-Markovian reward.** We are grateful to the reviewer for providing a comprehensive list of papers
on non-Markovian reward, and we are planning to add them into the related work section in the next version.

[Meta-Review · NeurIPS 2020]

This paper aims to extend the theory of RL to more general objective functions than just cumulative reward. For example, one may wish to optimize for maximum minimum return. The paper specifically studies _deterministic_ MDPs, which is a bit of a departure from standard RL setups, but seems reasonable for certain problems. Several conditions are established such that, when a generic objective satisfies all of them, there exists an algorithm that optimizes it to \epsilon optimality. Two learning algorithms are proposed: one specifically for symmetric norms; the other for generic objectives that satisfy the three aforementioned conditions. The reviews were overall very positive about the work. Supporting more general objective functions is a worthy, important problem to study, and the results are compelling. The primary criticism was about the focus on deterministic MDPs. Nonetheless, the reviewers ultimately weren't too bothered by this assumption. It will be interesting to see if the authors can extend this work to the more traditional stochastic setting. Another criticism is that the proposed algorithms are not validated empirically. I think experiments would have taken this paper to another level. There was some confusion over the fact that the algorithm's time complexity scales with the number of _multisets_ of cardinality k, not the number of length-k sequences. The reviewers and I am satisfied by the authors' response and thank them for clearing that up. While most reviewers agreed that the presentation is clear enough, there is some room for improvement. I encourage the authors to incorporate the reviewers' detailed feedback when revising.